# Thermoelectric coupling effect in BNT-BZT-$x$GaN pyroelectric ceramics for low-grade temperature-driven energy harvesting

Meng Shen [1,2] ✉, Kun Liu[1], Guanghui Zhang[1], Qifan Li[1], Guangzu Zhang [3], Qingfeng Zhang [1,4] ✉, Haibo Zhang [5], Shenglin Jiang[3], Yong Chen [1] ✉ & Kui Yao [2] ✉

Pyroelectric energy harvesting has received increasing attention due to its ability to convert low-grade waste heat into electricity. However, the low output energy density driven by low-grade temperature limits its practical applications. Here, we show a high-performance hybrid BNT-BZT-$x$GaN thermal energy harvesting system with environmentally friendly lead-free BNT-BZT pyroelectric matrix and high thermal conductivity GaN as dopant. The theoretical analysis of BNT-BZT and BNT-BZT-$x$GaN with $x = 0.1$ wt% suggests that the introduction of GaN facilitates the resonance vibration between Ga and Ti, O atoms, which not only contributes to the enhancement of the lattice heat conduction, but also improves the vibration of $TiO_6$ octahedra, resulting in simultaneous improvement of thermal conductivity and pyroelectric coefficient. Therefore, a thermoelectric coupling enhanced energy harvesting density of 80 μJ cm$^{-3}$ has been achieved in BNT-BZT-$x$GaN ceramics with $x = 0.1$ wt% driven by a temperature variation of 2 °C, at the optical load resistance of 600 MΩ.

Implementation of self-powered and battery-free sensors and other devices made from green materials composition is demanded for realizing energy and environmental sustainability[1,2]. Among them, the system capable of harvesting thermal waste is attracting special attention, because a large amount of energy produced by non-regeneration resources, such as coal, gas, and oil, is wasted as heat[3,4]. Eco-friendly methods to convert the waste heat into electricity can not only achieve reduction in carbon emissions, but also furnish renewable

and clear energy for self-powered sensors. Thermoelectric energy harvesting based on semiconductor materials has been widely explored due to their ability to turn spatial temperature gradients into electricity[5,6]. As a supplement of thermoelectrics, thermal energy harvesters based on pyroelectric effect in a ferroelectric material can convert waste heat of low temperature grade into electricity and work autonomously almost without requiring any external maintenance. Such pyroelectric energy harvesters are suitable for providing green

[1]Hubei Key Laboratory of Micro-Nanoelectronic Materials and Devices, Hubei Collaborative Innovation Center for Advanced Organic Chemical Materials, Ministry of Education Key Laboratory of Green Preparation and Application for Functional Materials, and School of Microelectronics, Hubei University, Wuhan 430062, China. [2]Institute of Materials Research and Engineering (IMRE), A*STAR (Agency for Science, Technology, and Research), Singapore 138634, Singapore. [3]School of Optical and Electronic Information and Wuhan National Laboratory for Optoelectronics, Huazhong University of Science and Technology, Wuhan, Hubei 430074, China. [4]Ministry of Education Key Laboratory of Green Preparation and Application for Functional Materials, Hubei Key Laboratory of Micro-Nanoelectronic Materials and Devices, Hubei Key Laboratory of Polymer Materials, School of Materials Science & Engineering, Hubei University, Wuhan 430062, China. [5]School of Materials Science and Engineering, State Key Laboratory of Material Processing and Die & Mould Technology, Huazhong University of Science and Technology, Wuhan 430074, China. ✉e-mail: sm@hubu.edu.cn; zhangqingfeng@hubu.edu.cn; chenyong@hubu.edu.cn; k-yao@imre.a-star.edu.sg

power for passive sensor network, medical health monitoring, and intelligent residential system[7–9].

The current of pyroelectric energy collector can be expressed as:

$$I_p = p \times A \times \frac{dT}{dt} \tag{1}$$

where $p$, $A$, and $dT/dt$ are the pyroelectric coefficient ($p$), the effective electrode area, and the rate of temperature change ($dT/dt$), respectively[8]. It is well known from the equation that the performance of pyroelectric energy harvester is proportional to pyroelectric coefficient ($p$) of the material and the rate of temperature change ($dT/dt$). There are different methods proposed for improving the pyroelectric energy harvesting performance, and some of them are achieved by increasing the $p$ or/and $dT/dt$[7,10–12]. For example, extensive attempts have been made to enhance $p$ by constructing morphotropic phase boundary (MPB) or polar nanoscale regions (PNRs) to improve the performance of pyroelectric energy harvesting[13,14]. Many efforts have also been made to increase the $dT/dt$ by building meshed electrodes or porous structures[15,16]. Simultaneous increase in $p$ and $dT/dt$ has been reported in PZT-based pyroelectric materials by introducing AlN or BN high thermal conductivity network to enlarge their thermal energy harvesting performance[17,18]. However, why the high thermal conductivity filler simultaneously leads to enhanced $p$ and $dT/dt$ in pyroelectric ceramics still remain an open theoretical question. This limits the further improvement in output energy density of the pyroelectric energy harvester for practical applications.

In this work, we design a hybrid system comprising lead-free BNT-BZT pyroelectric matrix and high thermal conductivity GaN as dopant to reveal the mechanism of the high thermal conductivity of fillers for enhancing thermal energy harvesting density and obtain environmentally friendly thermal energy harvester. The hybrid ceramic BNT-BZT-$x$GaN was fabricated by conventional solid-state reaction method. When $x = 0.1$ wt%, the peak pyroelectric coefficient reaches a large value of $850 \times 10^{-4}\,C\,m^{-2}\,K^{-1}$, which generates a short-circuit peak current of 0.12 µA and an open-circuit peak voltage of 58 V driven by a low-grade temperature difference of 2 °C. Our systematic study on the performance evolution with the GaN doping content, material microstructure, lattice vibrations and phonon structures, reveals the mechanism underlying the superior energy harvesting performance. As shown in Fig. 1, the introduction of GaN facilitates the resonance vibration between Ga and Ti, O atoms, leading to the enhancement of the lattice vibration of TiO$_6$ octahedra. This not only contributes to the enhancement of the lattice heat conduction, but also improves the spontaneous polarization, resulting in the simultaneous improvement of $dT/dt$ and $p$. Therefore, a high thermoelectrical energy harvesting density of 80 µJ cm$^{-3}$ at the optimal load resistance of 600 MΩ has been obtained in BNT-BZT-$x$GaN ceramic with $x = 0.1$ wt%. In addition, this thermoelectric coupling modulated energy harvesting system can charge the capacitor and control the capacitor to discharge and light up the LED bulb when connected with a capacitor. The understanding from this work provides a valuable guidance for designing pyroelectric materials with further improved thermal energy harvesting performance.

## Results and discussion
### Compositional heterogeneity and microstructure
To reveal microstructures and the compositional heterogeneity of the samples, the SEM morphology and average grain size are presented in Supplementary Fig. 1. As presented in Supplementary Fig. 1, all the samples have relatively compact grain morphology and the grain size firstly increases and then decreases slightly with the increase of GaN content. Meanwhile, the backscattered SEM images and the corresponding two-dimensional element mapping of BNT-BZT-$x$GaN with $x = 0.1$ wt% (as shown in Fig. 2) are compared with that of $x = 0$ (as shown

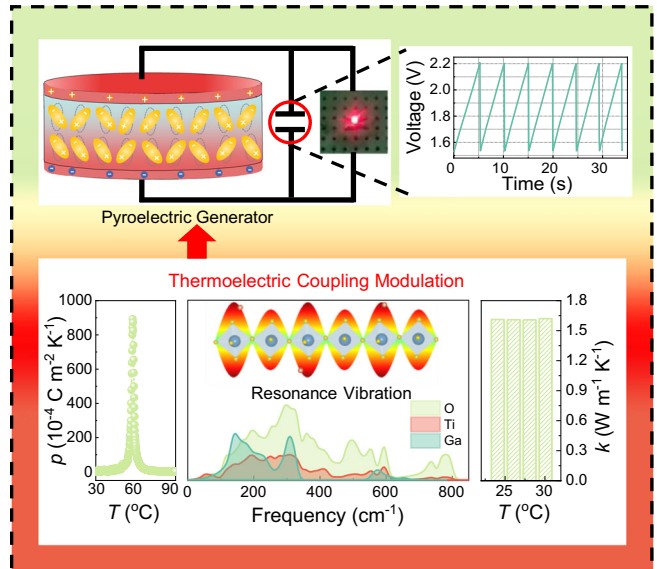

**Fig. 1 | Schematic diagram of a pyroelectric-based thermoelectrical energy harvesting system fabricated by BNT-BZT-0.1 wt% GaN ceramic.** The introduction of GaN facilitates the resonance vibration between Ga and Ti, O atoms and enhances the lattice vibration of TiO$_6$ octahedra, which not only contributes to the enhancement of the lattice heat conduction, but also improves the pyroelectric properties. This thermoelectric coupling modulated energy harvesting system can light up the LED bulb by storing the electricity in the capacitor. When the voltage of the capacitor reaches 2.2 V, the capacitor is triggered to discharge and light up the LED bulb.

in Supplementary Fig. 2). It can be observed that the elements of Na, Bi, and O are dispersed in the main area of the sample while the elements of Ba, Ti appear at the grain boundaries. It means that the introduction of GaN contributes to the agglomeration of BaTiO$_3$. For other lower-content elements, such as Zr, Ga, and N are evenly distributed in the ceramic matrix without clustering. Obviously, the elements Ga and N are uniformly distributed in the ceramics, which indicates that the GaN filler permeates into the BNT-BZT matrix, rather than gathering round the grain boundaries. As observed in EDS spectrum of Supplementary Fig. 3, all the elements of samples are detected, and the peaks of Ga and N are presented at 1.10 keV and 0.39 keV, manifesting the existence of GaN.

To analyze the lattice structures of BNT-BZT-$x$GaN samples, the XRD patterns and XRD Rietveld refinement are presented in Fig. 3a–d. As shown in Fig. 3a, the pure perovskite structures can be observed for all BNT-BZT-$x$GaN samples, suggesting that GaN has diffused into BNT-BZT lattice (as shown in Fig. 2d), being consistent with the analysis of EDS. The perovskite phase structure of BNT-BZT-$x$GaN ceramics can be confirmed by assessing the splitting peaks at around 40.0° and 46.5° as magnified in Fig. 3b. For the rhombohedral phase, only one peak can be observed at 46.5°, while the (200) peak turns into two peaks of tetragonal phase[19,20]. As shown in Fig. 3b, both the (100) peak at 40.0° and the (200) peak divide into two peaks, implying the coexistence of rhombohedral and tetragonal phase in BNT-BZT-$x$GaN system[21]. The rhombohedral phase dominates in the BNT-BZT-$x$GaN ceramic matrix because of the agglomeration of BaTiO$_3$ as shown in Fig. 2. The main crystal phase with $R3c$ space group (as shown in Fig. 3d) can be identified by the XRD Rietveld refinement as shown in Supplementary Fig. 4a–e. The reliability factors of weighted patterns ($R_{wp}$), patterns ($R_p$) and the goodness-of-fit indexes ($\chi^2$) are under 7.5%, 5.5%, and 5.5%, respectively, indicating that the crystal structure mode is valid and the refinement result is reliable. The lattice parameters ($a = b = c$, $\alpha = \beta = \gamma$ and volume) and the distance of Ti-O are obtained as shown in Fig. 3c. The angles ($\alpha = \beta = \gamma$) tend to decrease, while both the lattice parameters ($a$, $b$, $c$) and volumes first increase and then decrease with

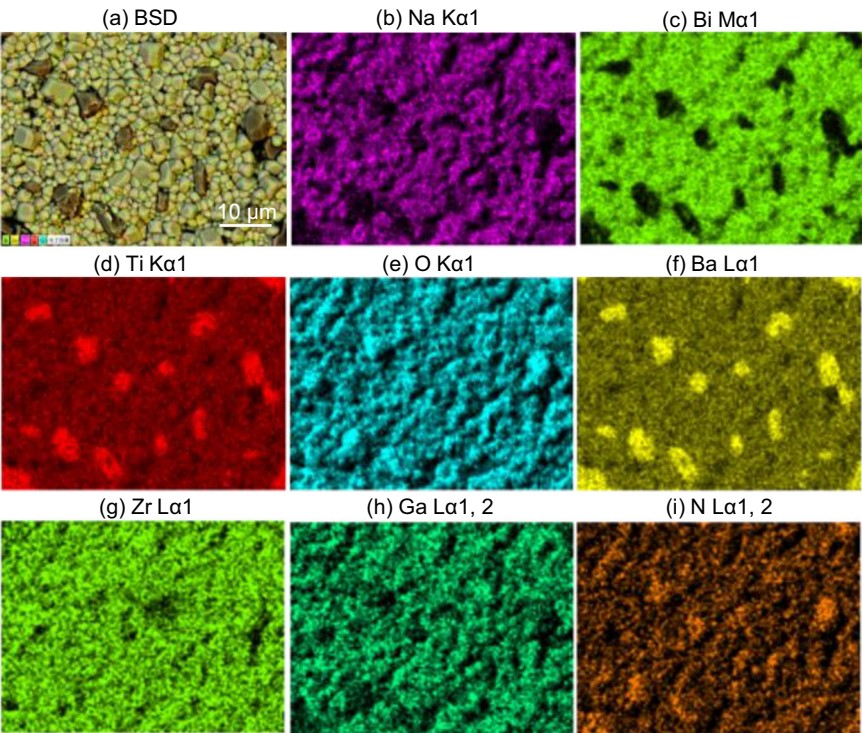

**Fig. 2 | Compositional heterogeneity and microstructure measured by FE-SEM and EDS. a–i** The backscattering diffraction (BSD) image and the corresponding elemental distribution of BNT-BZT-$x$GaN with $x$ = 0.1 wt%.

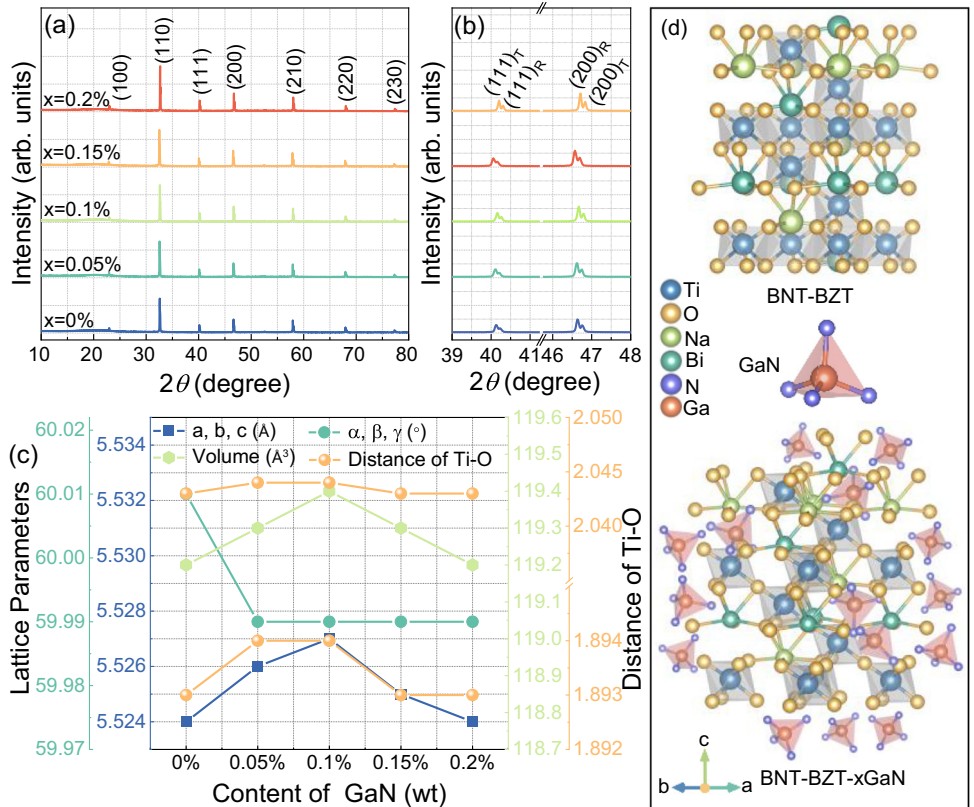

**Fig. 3 | Crystal structure measured by XRD and the Rietveld refined data calculated by the GSAS-EXPGUI software. a** The XRD patterns of BNT-BZT-$x$GaN ceramics with various contents of GaN, **b** the magnified XRD peaks at ~40.0° and 46.5° for BNT-BZT-$x$GaN samples with $x$ = 0–0.2 wt%, **c** the Rietveld refined lattice parameters ($a$, $b$, $c$, $\alpha$, $\beta$, $\gamma$) and distance of Ti-O for BNT-BZT-$x$GaN samples with various contents of GaN, and **d** the structural images of BNT-BZT and BNT-BZT-$x$GaN samples, respectively.

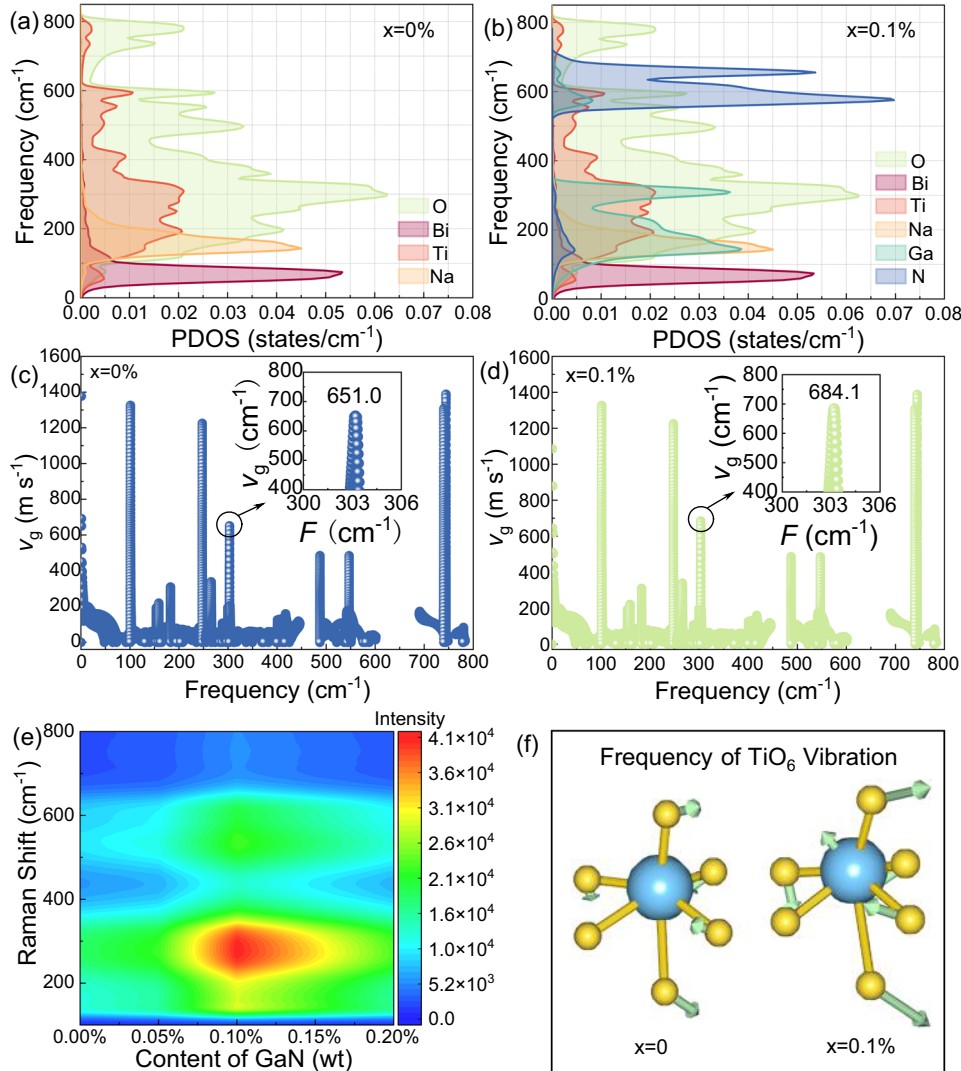

**Fig. 4 | The theoretical calculation of lattice vibrations and phonon structure by CASTEP.** The phonon density of state (PDOS) of **a** BNT-BZT and **b** BNT-BZT-$x$GaN with $x = 0.1$ wt%, and the phonon group velocity of **c** BNT-BZT and **d** BNT-BZT-$x$GaN with $x = 0.1$%. **e** The Raman scattering spectra of BNT-BZT-$x$GaN with $x = 0 - 0.2$ wt% measured by laser Raman spectrometer. **f** The vibration frequency of Ti-O for BNT-BZT and BNT-BZT-$x$GaN with $x = 0.1$ wt%, respectively.

addition of GaN. In addition, the similar tendency is also observed in the Ti-O bond distances of BNT-BZT-$x$GaN samples. As the GaN content increases to 0.1 wt%, the Ti-O (1) and Ti-O (2) bond lengths reach the maximum value of 1.894 Å and 2.044 Å.

## Lattice vibrations and phonon structure

In BNT-based dielectric solids, lattice vibrations dominate the heat transport and a phonon refers to the quantum of atomic vibrational energy. The thermal transport in solids is regarded as the diffusion of phonons actuated by a heat gradient[22]. To analyze the relation between lattice vibration and phonon structure of BNT-BZT-$x$GaN samples, the calculated phonon-dispersion curves along the high-symmetry lines of Brillouin zone of BNT-BZT and BNT-BZT-$x$GaN with $x = 0.1$ wt% are presented in Supplementary Fig. 5. There are three low-frequency acoustic modes observed near the Γ-point and a lack of imaginary line in phonon band structures proves the dynamical stability for both BNT-BZT and BNT-BZT-$x$GaN with $x = 0.1$ wt% samples[23,24]. Meanwhile, the dispersion curves show some optical modes above 80 cm$^{-1}$ and a band gap between 600 and 680 cm$^{-1}$. The optical modes below 800 cm$^{-1}$ and the acoustic modes determine the lattice thermal conductivity, playing an important role in heat transport. To find out intrinsic nature of lattice thermal conduction, the phonon density of

state (PDOS) of BNT-BZT and BNT-BZT-$x$GaN with $x = 0.1$ wt% is presented in Fig. 4a, b. As shown in Fig. 4a, b, there are different shadowed areas representing Bi, Na, Ti, O, Ga, and N atom phonon DOS, respectively. The atoms of BNT-based matrix mainly possess phonon distribution within the low-frequency part blew 800 cm$^{-1}$. This means that atoms for BNT-based matrix vibrate at the phonon modes below 800 cm$^{-1}$ and conduct heat, corresponding to the phonon band dispersion as shown in Supplementary Fig. 5b. In addition, the phonon distribution of Ga atom is located in the region between 10 and 375 cm$^{-1}$, which has overlapped with the atom phonon DOS of Na, Ti, and O. These overlapped phonon distributions stand for the resonance vibration between Ga and Na, Ti, O atoms, which contributes to the enhancement of the lattice vibration of BNT-BZT. The enhancement of the lattice vibration of BNT-BZT not only gives rise to the enhancement of the lattice heat conduction, but also improves the spontaneous polarization of the samples. As the spontaneous polarization of BNT-based ceramics is prevailingly derived from the movement of Ti and O atoms, the resonance vibration among Ga and Ti, O atoms (as shown in Fig. 4b) improves the vibration amplitude and frequency of Ti-O (as shown in Fig. 4f and Supplementary Movie 1), which can be confirmed by the group velocity of BNT-BZT and BNT-BZT-$x$GaN, with $x = 0.1$ wt% as shown in Fig. 4c, d. The phonon

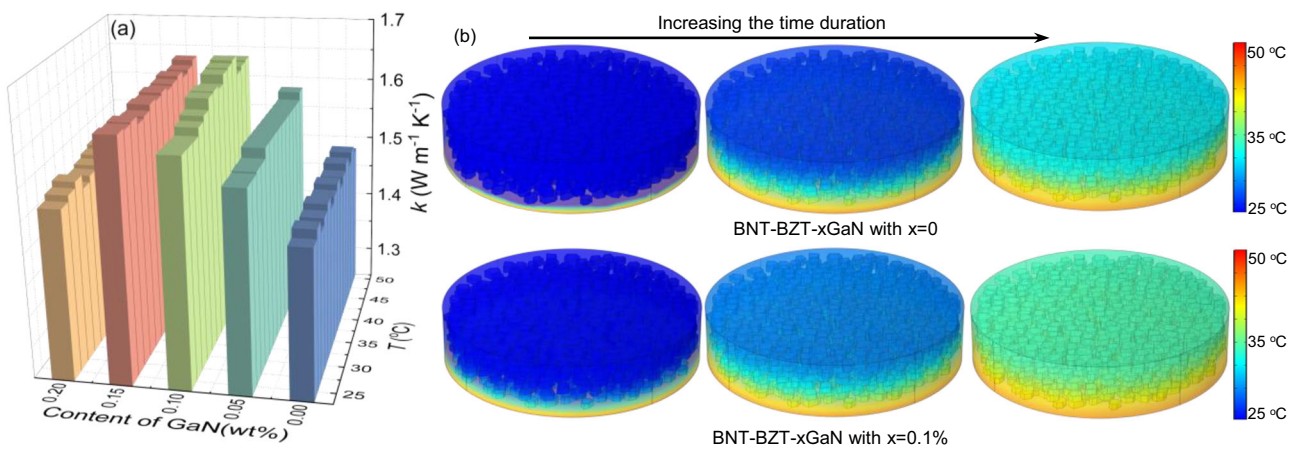

**Fig. 5 | Thermal transport. a** The temperature-dependent thermal conductivity of BNT-BZT-$x$GaN ceramics calculated by the equation $k = \alpha\rho C_p$, and **b** the simulation of temperature distributions for the pristine and BNT-BZT-$x$GaN sample with $x = 0.1$ wt% by COMSOL Multiphysics software at the same time duration.

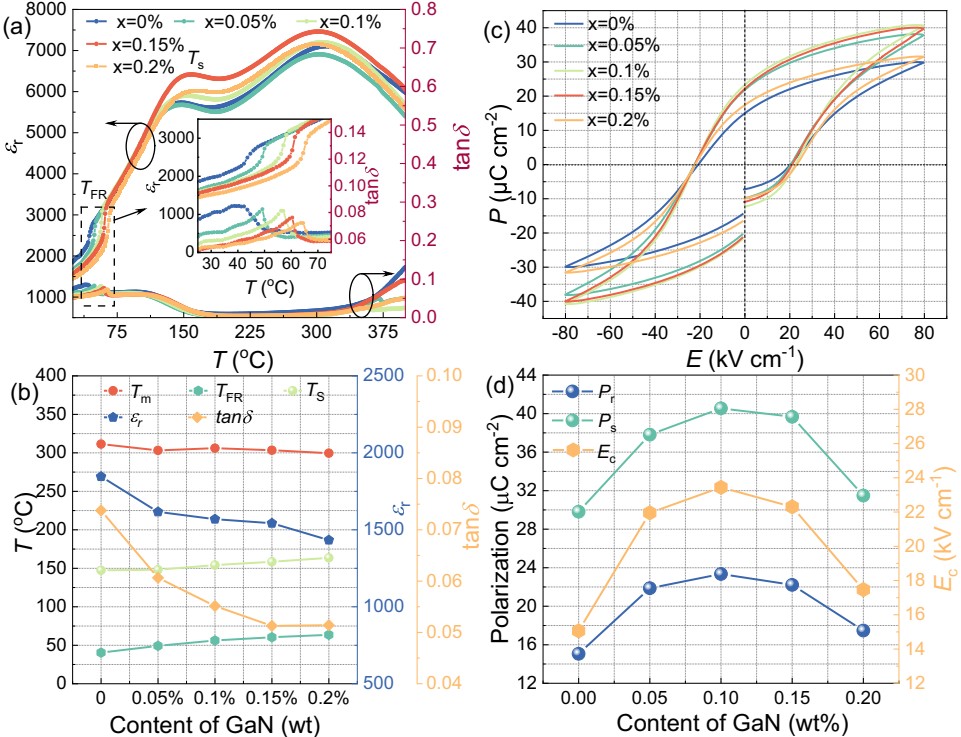

**Fig. 6 | Dielectric and ferroelectric properties measured with dielectric properties testing system and ferroelectric analyzer system. a** The temperature-dependent dielectric constant ($\varepsilon_r$) and loss ($tan\delta$) for BNT-BZT-$x$GaN samples with various content of GaN in the temperature range of 25–400 °C, **b** the temperature of ferroelectric-paraelectric phase transition ($T_m$), ergodic relaxor-ferroelectric state ($T_s$), non-ergodic to ergodic relaxor state ($T_{FR}$), $\varepsilon_r$ and $tan\delta$ of BNT-BZT-$x$GaN samples with $x = 0$–0.2 wt%, **c** the $P$-$E$ hysteresis loops, and **d** the saturation polarization ($P_s$), remnant polarization ($P_r$) and coercive electric field ($E_c$) of BNT-BZT-$x$GaN samples with various content of GaN at room temperature.

group velocity of BNT-BZT-$x$GaN with $x = 0.1$% at ~300 cm$^{-1}$ is 684.1 m s$^{-1}$, roughly corresponding to the Ti-O vibration (shown in Fig. 4e and Supplementary Fig. 6). This value is higher than 651.0 m s$^{-1}$ of pristine BNT-BZT sample as shown in Fig. 4c. Meanwhile, as shown in Fig. 4e and Supplementary Fig. 6, the intensity of Raman spectrum between 10 and 375 cm$^{-1}$ for BNT-BZT-$x$GaN with $x = 0.1$% is significantly stronger than that of pure BNT-BZT sample, further confirming the resonance vibration among Ga and Ti, O atoms. These results show that the introduction of GaN achieves the simultaneous improvement of thermal conductivity and spontaneous polarization, which can be verified by the measurement of heat conductivity and electrical properties as shown in Figs. 5 and 6, respectively.

## Thermal conductivity and heat transport
Based on the kinetic theory of gases and Debye's specific heat theory, the lattice thermal conductivity of a crystalline solid can be expressed as: $\kappa = \frac{1}{3}C_v v_g^2 \tau$, where $C_v$ is the specific heat per unit volume, $v_g$ is the phonon group velocity, $\tau$ is the relaxation time[25]. According to the equation, the thermal conductivity is proportional to the square of the phonon group velocity. As discussed in Fig. 4c, d, with the modification of GaN, the phonon group velocity of BNT-BZT-$x$GaN with $x = 0.1$ wt% is higher than that of pristine BNT-BZT sample. Theoretically, the thermal conductivity of BNT-BZT-$x$GaN with $x = 0.1$ wt% is correspondingly higher than that of pristine BNT-BZT sample. To evaluate the actual heat conductivity, Fig. 5a and Supplementary Fig. 7 give the

temperature-dependent thermal parameters of BNT-BZT-$x$GaN ceramics with different amount of GaN. As shown in Fig. 5a, while the thermal conductivity does not vary much with an increase in temperature, the introduction of GaN contributes to an increase in the thermal conductivity from 1.48 W m$^{-1}$ K$^{-1}$ to 1.61 W m$^{-1}$ K$^{-1}$ when the GaN content increases from 0 to 0.1 wt% at room temperature. This may be contributed to the resonance vibration and enhancement in the phonon group velocity resulted from introduction of GaN. As the GaN content continues to increase to 0.2 wt%, the thermal conductivity of BNT-BZT-$x$GaN decreases to 1.52 W m$^{-1}$ K$^{-1}$. The decrease in thermal conductivity may be attributed to the decrease in mean-free-paths due to phonon-boundary scattering. As we know, heat is carried by phonons with a broad distribution of mean-free-paths and the thermal conductivity is proportional to mean-free-paths. As the GaN content further increases to 0.2 wt%, the grain size of BNT-BZT-$x$GaN ceramics decreases to 1.57 μm and introduces more grain boundaries. With the introduction of boundaries, phonons with long mean-free paths interact with the boundaries and reduce their mean-free paths, which narrow the distribution of mean-free-paths. Therefore, the thermal conductivity of BNT-BZT-$x$GaN decreases to 1.52 W m$^{-1}$ K$^{-1}$ when the GaN content further increases to 0.2 wt%. In addition, the temperature distributions of the pristine and BNT-BZT-$x$GaN sample with $x = 0.1$ wt% calculated by COMSOL Multiphysics software at the same time duration are presented in Fig. 5b. It can be observed that BNT-BZT-$x$GaN sample with $x = 0.1$ wt% is able to conduct heat faster for the same thermal input in comparison with the pure ceramic. Namely, the construction of BNT-BZT-$x$GaN hybrid ceramic can be an effective method to improve the heat transfer performance compared to the pure BNT-BZT ceramic samples.

## Dielectric and ferroelectric properties

Figure 6a and Supplementary Fig. 8 present the dielectric constant ($\varepsilon_r$) and loss ($tan\delta$) with temperature range from 25 to 400 °C for polarized BNT-BZT-$x$GaN samples. As shown in Fig. 6a and Supplementary Fig. 8, three dielectric peaks can be observed in dielectric constant curves for all samples, which correspond to ferroelectric to paraelectric phase transition ($T_m$) at ~300 °C, ergodic relaxor to ferroelectric state ($T_s$) at ~150 °C, and non-ergodic to ergodic relaxor state ($T_{FR}$) near ambient temperature, respectively[26]. The $T_m$ decreases slightly while the $T_s$ and $T_{FR}$ increase appropriately with the increase in GaN content as presented in Fig. 6b. In particular, the temperature of non-ergodic to ergodic relaxor state ($T_{FR}$) for BNT-BZT-$x$GaN samples shifts from 40 to 65 °C with increasing GaN, which is corresponding to the peak pyroelectric temperature as shown in Fig. 7a. In addition, the dielectric constant ($\varepsilon_r$) and loss ($tan\delta$) at corresponding ambient temperature range decrease with the increase in GaN content, especially in room temperature from ~1750 to ~1400 and from ~0.074 to ~0.05, respectively. Because the pyroelectric energy harvesting figure of merit is inversely proportional to the dielectric constant, the decrease in dielectric constant ($\varepsilon_r$) facilitates the improvement of pyroelectric energy harvesting performance. The polarization-electric field ($P$-$E$) hysteresis loops of BNT-BZT-$x$GaN samples with various content of GaN at room temperature are presented in Fig. 6c. All the samples exhibit typical ferroelectric hysteresis loops. Both saturation polarization ($P_s$) and remnant polarization ($P_r$) first increase and then decrease with increasing GaN content, as the GaN content increases to 0.1 wt%, both the $P_s$ and $P_r$ gradually increase to the maximum value of 40 μC cm$^{-2}$ and 23 μC cm$^{-2}$, respectively, which is attributed to the enhancement of Ti-O vibration as discussed in Fig. 4.

## Pyroelectric effect and mechanism of pyroelectric energy harvesting

Figure 7a presents the temperature-dependent pyroelectric coefficient of BNT-BZT-$x$GaN ceramics with different content of GaN in the temperature range of 20-90 °C. It can be observed from Fig. 7a that the peak pyroelectric coefficient of BNT-BZT-$x$GaN samples shows a similar trend as that of saturation polarization and remnant polarization. The peak pyroelectric coefficient firstly increases and then decreases with the addition of GaN, and reaches the maximum value of 850 × 10$^{-4}$ C m$^{-2}$ K$^{-1}$ when the content of GaN increases to 0.1 wt%. Meanwhile, the peak position gradually shifts to higher temperature with the increase in GaN, in accordance with the temperature of non-ergodic to ergodic relaxor state as shown in Fig. 7b. The mechanisms of pyroelectric energy harvesting for BNT-BZT and BNT-BZT-$x$GaN with $x = 0.1$ wt% are illustrated in Figs. 7c, d, respectively. At a steady state ($dT/dt = 0$), for BNT-BZT samples, the polarized electric dipoles reorient along the applied electric field and oscillate randomly to reach an equilibrium condition, concomitantly obtaining the attracted charges on both surfaces of samples as shown in Fig. 7c. For BNT-BZT-$x$GaN with $x = 0.1$ wt%, there is an enhanced dipole pinning effect compared with pure ceramics due to the interaction of two types of dipoles in BNT-BZT-$x$GaN ceramics. One is electric dipoles originated from relative displacement between B-site Ti ions and O ions as shown in Fig. 7c, d and the other one is defect dipoles formed by various lattice defects. During poling, both the lattice displacement electric dipoles and defect dipoles reorient along the applied electric field, when the electric field is removed, both the lattice displacement electric dipoles and defect dipoles oscillate to reach new equilibrium conditions. The defect dipoles usually have the low migration speed of defect. Meanwhile, the defect dipoles interact with lattice displacement electric dipoles acting as pinning points for the electric dipoles motion and thus providing pinning effects in electric dipoles activities. These can be confirmed by the shift of ferroelectric hysteresis ($P$-$E$) loops along the electric field axis (defined as internal bias field) and constriction of the $P$-$E$ loops as shown in Supplementary Fig. 9. Therefore, the poled electric dipoles of BNT-BZT-$x$GaN with $x = 0.1$ wt% are absorbed a greater quantity of opposite electrical charge on both sides of samples. When the sample is heated, the electric dipoles in BNT-BZT ceramics oscillate within a large degree of alignment, leading to the decrease in spontaneous polarization and absorb electrical charges, and thus currents flow in the external circuit. With the introduction of GaN, heat transfer is faster and the vibration of Ti-O is enhanced in BNT-BZT-$x$GaN with $x = 0.1$ wt% owing to the improved lattice thermal conductivity, which contributes to dipoles' oscillation in even larger degree and larger currents in the circuits as illustrated in Fig. 7d.

## Pyroelectric energy harvesting

Figure 8 presents the energy harvesting performance of BNT-BZT-$x$GaN samples in response to temperature change between 25 and 50 °C. The corresponding time dependent temperature variation curves are exhibited in Fig. 8a. It can be seen that the thermal distribution is a combination of continuous rising trend with periodic temperature fluctuation of ~2 °C every 10 s. The short-circuit current and open-circuit voltage of BNT-BZT-$x$GaN ceramic induced by the low temperature variation of 2 °C are shown in Fig. 8b, c. As shown in Fig. 8b−c, both short-circuit current and open-circuit voltage increase first and then decrease with the increase in GaN content. As the content of GaN reaches 0.1 wt%, the BNT-BZT-$x$GaN sample with $x = 0.1$ wt % can generate a short-circuit peak current of 0.12 μA, or an open-circuit peak voltage of 58 V. Meanwhile, short-circuit current and open-circuit voltage of BNT-BZT-$x$GaN ceramic at different temperature variation (ΔT) of 1 °C, 3 °C, and 4 °C are shown in Supplementary Fig. 10. As shown in Supplementary Fig. 10a−c, both short-circuit current and open-circuit voltage show similar trend of increasing first and then decreasing with further increase in GaN content, and the short-circuit current and open-circuit voltage reach the maximum value in BNT-BZT-$x$GaN ceramic with $x = 0.1$ wt% for all the cases of ΔT of 1 °C, 3 °C, or 4 °C. In addition, Supplementary Fig. 10d, e compare the peak short-circuit currents and peak open-circuit voltages of BNT-

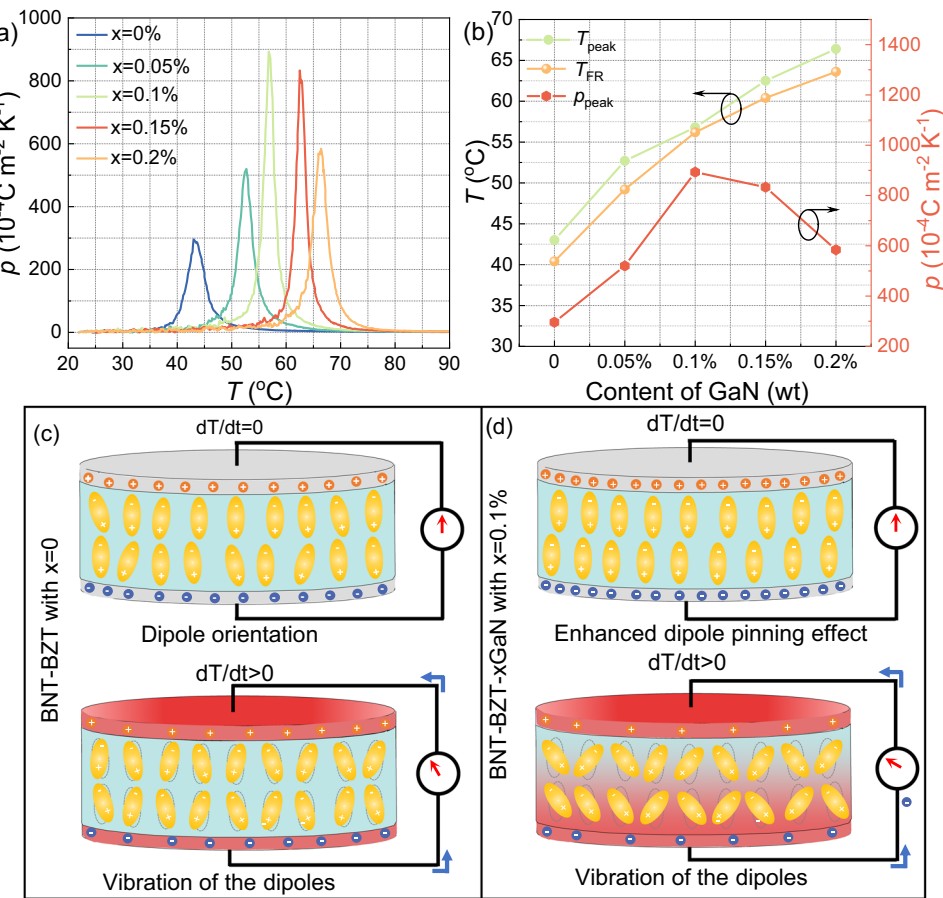

**Fig. 7 | Pyroelectric property measured by Pyroelectric Properties Testing System and schematic illustration for the mechanisms of pyroelectric energy harvesting. a** The temperature-dependent pyroelectric coefficient of BNT-BZT-$x$GaN ceramics with different content of GaN, **b** the temperature at peak pyroelectric coefficient ($T_{peak}$), the temperature at non-ergodic to ergodic relaxor state ($T_{FR}$) and peak pyroelectric coefficient ($p_{peak}$) of BNT-BZT-$x$GaN ceramics with $x = 0$–0.2 wt%. **c**, **d** The illustration for the mechanisms of pyroelectric energy harvesting for BNT-BZT and BNT-BZT-$x$GaN with $x = 0.1$ wt%, respectively (At a steady state ($dT/dt = 0$), in comparison with BNT-BZT sample, the poled electric dipoles of BNT-BZT-$x$GaN with $x = 0.1$ wt% ceramic absorb a greater quantity of opposite electrical charge on both sides of samples due to the enhanced dipole pinning effect, providing the potential for giant pyroelectric effect. When $dT/dt' 0$, the electric dipoles in BNT-BZT-$x$GaN with $x = 0.1$ wt% ceramic oscillate within a larger degree of alignment, leading to larger currents in the circuits compared with that of BNT-BZT samples.).

BZT-$x$GaN ceramics with ΔT at 1 °C, 2 °C, 3 °C, and 4 °C, respectively. It can be seen from Supplementary Fig. 10d, e that both the peak short-circuit current and peak open-circuit voltage increase with ΔT. The improvement in pyroelectric short-circuit current and open-circuit voltage are mainly owing to the thermoelectrical coupling effect in BNT-BZT-$x$GaN sample with $x = 0.1$ wt% modulated by GaN. In addition, the pyroelectric current and voltage of BNT-BZT-$x$GaN sample with $x = 0.1$ wt% obtained at different load resistance are presented in Fig. 8d, e. As shown in Fig. 8d, e and Supplementary Fig. 11, the pyroelectric current is approximately equal to the short-circuit current and pyroelectric voltage is approximately equal to zero when the load resistance is lower than 1 MΩ. When the load resistance is larger than 1 MΩ, the pyroelectric current decreases with the increase in load resistance while the pyroelectric voltage increase with the increase in load resistance. Meanwhile, the output energy density of BNT-BZT-$x$GaN ceramics with $x = 0.1$ wt% has been investigated by measuring the pyroelectric current and voltage with a range of load resistance and calculation based on the integral formula $W = \int_{t=0}^{t=T} \frac{I \times U}{S \times d} dt$, where $I$, $U$, $T$, $S$, and $d$ are pyroelectric current, voltage, one cycle of 10 s, sample area and thickness, respectively. Figure 8f shows the load resistance dependent energy density curves of BNT-BZT-$x$GaN ceramics with $x = 0.1$ wt% induced by the low temperature fluctuation of 2 °C. As shown in Fig. 8f, the output energy density of a period is approximately equal to zero when the load resistance is lower than 1 MΩ, When the load resistance is larger than 1 MΩ, the output energy density firstly increases and then decreases with the increase in load resistance and reaches the maximum value of 80 μJ cm$^{-3}$ at the load resistance of 600 MΩ. In addition, Fig. 8g presents the electrical circuit diagram of the pyroelectric energy harvesting device where a 4.8 μF load capacitor and BNT-BZT-0.1 wt%GaN sample are connected by a phase-controlled rectifier. The pyroelectric samples generate alternating current driven by a low thermal gradient, which is converted by the phase-controlled rectifier to direct current and stores the electricity in the capacitor. When the voltage of the capacitor reaches 2.2 V, the capacitor is triggered to discharge and light up the LED bulb. The charging-discharging voltage variation curves of the capacitor are presented in Supplementary Fig. 12. Also, the practical applicability of the thermal energy harvesting device induced by low temperature fluctuation is illustrated in Supplementary Movie 2. The BNT-BZT-$x$GaN ceramics with $x = 0.1$ wt% exhibit superior pyroelectric energy density driven by unit temperature gradient in comparison with state-of-the-art pyroelectric energy harvesting systems (as shown in Table 1), which have great potential application in low temperature driven thermal energy harvesting devices[16,27–39].

In summary, BNT-BZT pyroelectric ceramics with embedded high thermal conductive GaN are designed and fabricated by conventional

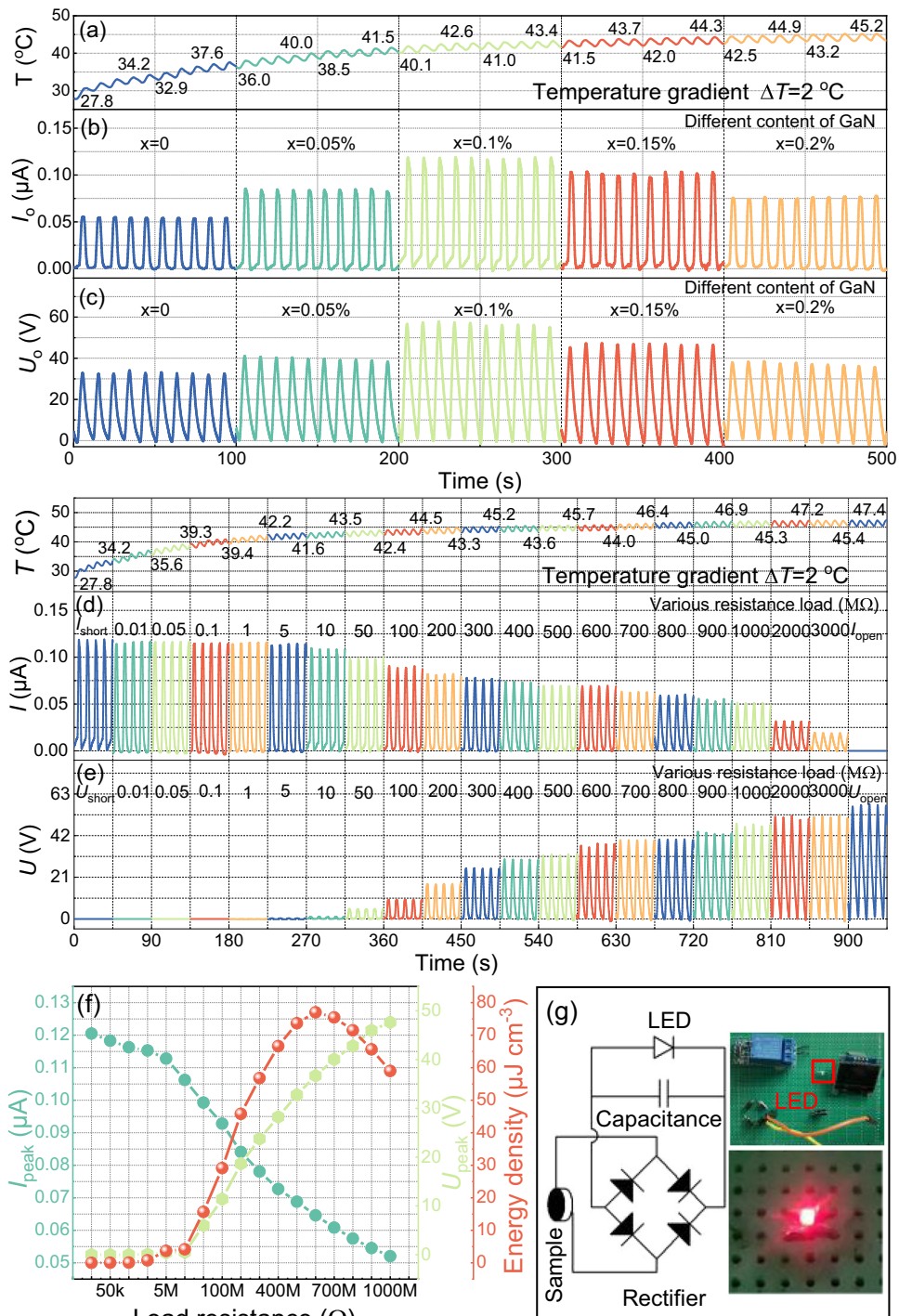

**Fig. 8 | Pyroelectric energy harvesting experimental results measured by the Thermal Energy Harvesting Testing System. a** The time-dependent temperature variation curves of samples. **b**, **c** The short-circuit current and open-circuit voltage of BNT-BZT-$x$GaN ceramics with various content of GaN. **d**, **e** The pyroelectric current and voltage obtained at different load resistance from 0–3000 MΩ. **f** The peak current, peak voltage, and output energy density (calculated by the equation $W = \int_{t=0}^{t=T} \frac{I \times U}{S \times d} dt$) of BNT-BZT-$x$GaN ceramic with $x = 0.1$ wt%. **g** The electrical circuit diagram of the pyroelectric energy harvesting device and the photograph of the LED lit up by the device.

solid-state reaction method to realize high pyroelectric energy harvesting performance. The hybrid BNT-BZT-$x$GaN ceramic exhibited a peak pyroelectric coefficient of $850 \times 10^{-4}$ C m$^{-2}$ K$^{-1}$, when $x = 0.1$ wt%, generating a short-circuit peak current of 0.12 μA or an open-circuit peak voltage of 58 V driven by a low temperature change of 2 °C, which exhibits superior pyroelectric energy density driven by unit temperature fluctuation in comparison with state-of-the-art pyroelectric energy harvesting systems. With a systematic study on the

thermoelectrical output, microstructure, lattice vibrations and phonon characteristics of the materials with the varied GaN doping content, the mechanism underlying the enhancement in the energy harvesting performance has been revealed. The introduction of GaN facilitates the resonance vibration with interactions among Ga and Ti, O atoms, which contributes to the enhancement of not only the lattice heat conduction, but also the vibration of Ti-O. This results in the simultaneous improvement of thermal conductivity and spontaneous

**Table 1 | Comparison of pyroelectric energy harvesting performance of BNT-BZT-xGaN ceramic with that of other state-of-the-art pyroelectric energy harvesting systems driven by unit temperature change**

| Materials | ΔT (K) | $p$ ($10^{-4}$ C m$^{-2}$ K$^{-1}$) | $U_o$ (V K$^{-1}$) | $I_o$ (μA K$^{-1}$) | Energy density for volume (μJ cm$^{-3}$ K$^{-1}$) | Power density for area (μw cm$^{-2}$ K$^{-1}$) | Ref. |
|---|---|---|---|---|---|---|---|
| ITO/BNT-BZT/Ag | 30.5 | 5.0 | 1.67 | 0.003 | ~ | ~ | 27 |
| BZT/BCT/STO | 11.8 | 34.3 | ~ | 0.11 | ~ | ~ | 28 |
| I$^-$/I$^{3-}$/MC/KCl (TGC) | 15.0 | ~ | 0.01 | ~ | 5.34 | ~ | 29 |
| PPGO | 110.0 | ~ | 0.02 | 0.10 | ~ | 0.17 | 30 |
| PMN-PMS-PZT: xCNT | 20.0 | 43.3 | 0.67 | ~ | ~ | ~ | 16 |
| HfO$_2$/Hf$_{0.5}$Zr$_{0.5}$O$_2$:La | 160.0 | 0.7 | ~ | ~ | ~ | ~ | 31 |
| Ag/PVDF/Ag | 4.0 | ~ | 15 | ~ | ~ | ~ | 32 |
| PVDF/BST/BN | 70.0 | ~ | 0.43 | 0.27 | ~ | 0.09 | 33 |
| SnS:Na | 3.9 | ~ | 0.001 | 24.51 | ~ | ~ | 34 |
| PEDOT: Tos | 13.8 | ~ | 5.41 | 0.001 | ~ | 0.03 | 35 |
| WKF-based PTM | 80.0 | ~ | ~ | ~ | ~ | 0.002 | 36 |
| Au@rGO-PEI | 29.0 | ~ | 4.14 | 0.072 | ~ | 0.009 | 37 |
| Al/PVDF/Al | 5.0 | 17.9 | 9.60 | 0.005 | 0.23 | ~ | 38 |
| CH$_3$NH$_3$PbI$_3$/PVDF | 38.0 | ~ | 0.001 | ~ | ~ | 0.003 | 39 |
| BNT-BZT-GaN | 2.0 | 850.0 | 29.0 | 0.06 | 40 | ~ | This work |

*ΔT temperature variation, p pyroelectric coefficient, $U_o$ open-circuit voltage generated at ΔT = 1°C, $I_o$ short-circuit current generated at ΔT = 1°C, ITO/BNT-BZT/Ag ITO/0.94(Bi$_{0.5}$Na$_{0.5}$)TiO$_3$-0.06Ba(Zr$_{0.25}$Ti$_{0.75}$)O$_3$/Ag, BZT/BCT/STO BaZr$_{0.2}$Ti$_{0.8}$O$_3$/Ba$_{0.7}$Ca$_{0.3}$TiO$_3$/ Ba$_{0.7}$Ca$_{0.3}$TiO$_3$, I$^-$/I$^{3-}$/MC/KCl (TGC) I–/I3– redox couple/methylcellulose/KCl (thermogalvanic cells), PPGO poly (3,4-ethylenedioxythiophene) poly (styrene sulfonate) and graphene oxide (PPGO), PMN-PMS-PZT: xCNT Pb[(Mn$_{1/3}$Nb$_{2/3}$)1/2(Mn$_{1/3}$Sb$_{2/3}$)1/2]$_{0.04}$(Zr$_{0.95}$Ti$_{0.05}$)$_{0.96}$O$_3$:x carbon nanotubes, Ag/PVDF/Ag Ag/Poly(vinylidene fluoride)/Ag, PVDF/BST/BN poly(vinylidene fluoridetrifluoroethylene-chlorofluoroethylene)/barium strontium titanate/boronnitride, PEDOT Tos tosylate-doped poly(3,4-ethylenedioxythiophene), WKF-based PTM Kevlar fiber based personal thermal management, Au@rGO-PEI Au@polyethylenimine modified graphene oxide, Al/PVDF/Al Al/Poly(vinylidene fluoride)/Al, CH3NH3PbI3/PVDF methylammonium lead iodide/poly(vinylidene fluoride).*

polarization, and hence the significant improvement in the thermoelectric coupling effect. A high energy harvesting density of 80 μJ cm$^{-3}$ at the optimal load resistance of 600 MΩ has been achieved in the BNT-BZT-xGaN ceramic with x = 0.1 wt%. The theoretical understanding provides a guidance to establish new principles for designing pyroelectric materials with further improved thermal energy harvesting performance.

## Methods
### Sample preparation
The solid-state reaction method was used to prepare 0.94Na$_{0.5}$Bi$_{0.5}$TiO$_3$-0.06BaTi$_{0.75}$Zr$_{0.25}$O$_3$ ceramic samples. According to the chemical formula, reagent grade metal oxide Bi$_2$O$_3$ (99%, Sinopharm), Na$_2$CO$_3$ (99.8%, Sinopharm), BaCO$_3$ (99%, Sinopharm), TiO$_2$ (98%, Sinopharm), and ZrO$_2$ (99%, Sinopharm) were ball milled in ethanol with zirconia for 10-18 h. After drying and calcination at 800-900 °C for 2 h, the mixtures were ball milled for 10-18 h again, then dried and mixed with commercial GaN nanoparticle according to the chemical formula (1-x) 0.94Na$_{0.5}$Bi$_{0.5}$TiO$_3$-0.06BaTi$_{0.75}$Zr$_{0.25}$O$_3$-xGaN (x = 0, 0.05, 0.1, 0.15, 0.2 wt%) (BNT-BZT-xGaN). After granulated with 5% polyvinyl alcohol, the mixed powders were pressed into a disk in a prototype abrasive tool at 5-10 MPa. Finally, sintering was carried out in aluminum oxide crucibles at 1130-1160 °C for 2 h with decreasing temperature rate of 3 °C min$^{-1}$.

### Compositional heterogeneity and microstructure characterization
Field emission scanning electron microscopy (FE-SEM) (Zeiss Geminisem 300) and energy-dispersive X-ray spectroscope (EDS) (Oxford instruments X-MaxN SN 78861) were employed to analyze the compositional heterogeneity and microstructures of the samples. The X-ray diffraction (XRD) (D8 Advance, Bruker, Germany) was utilized to characterize crystal structure. The refined data were obtained through Rietveld method on the basis of the slow-scan XRD data by using the GSAS-EXPGUI software and the Na$_{0.5}$Bi$_{0.5}$TiO$_3$ (space group R3c) CIF was chosen as the initial model[40,41].

### Calculation of phonon-dispersion, DOS and phonon group velocity
The Raman scattering spectra of the samples were measured by laser Raman spectrometer (LabRAM HR Evolution, Horiba, Japan). To analyze the phonon spectra, phonon density of state (PDOS), and phonon group velocity, first-principles calculations were performed within the Cambridge Serial Total Energy Package (CASTEP). The PDOS is calculated by the equation: $g^j(\omega, \hat{\mathbf{n}}) = \frac{1}{N}\sum_\lambda \delta(\omega - \omega_\lambda)\left|\hat{\mathbf{n}} \cdot e_\lambda^j\right|^2$, where N is the number of unit cells, j is the atom indices, δ is the Sommerfeld constant, ω is specified arbitrary using DOS-RANGE, $\omega_\lambda$ is phonon frequency, $\hat{\mathbf{n}}$ is the unit projection direction vector, and **e** is the polarization vector[40,42]. The phonon group velocity is calculated by the formula: $v_g(\mathbf{q}\nu) = \frac{1}{2\omega(\mathbf{q}\nu)}\left\langle e(\mathbf{q}\nu)\frac{\partial D(\mathbf{q})}{\partial \mathbf{q}}e(\mathbf{q}\nu)\right\rangle$, where **q** is the wave vector, $\nu$ is the index of phonon mode, $\omega$ is the phonon frequency, and D(q) is dynamical matrix. The optimized lattice parameters and IFCs are obtained from the total energy calculations by using plane augmented wave method based on density functional theory with Vienna ab initio simulation package (VASP)[43]. The electronic wave functions are expanded in a plane wave basis set with an energy cutoff of 680 eV. Such a high cutoff was found necessary to converge to the phonon-dispersion curves. The Brillouin zone was sampled with 6 × 6 × 6 k-points Monkhorst-Pack meshes for primitive cells of R3c phases of BNT-BZT-xGaN[44,45].

### Measurement of thermal parameters
The thermal conductivity (k) was calculated according to the equation $k = \alpha \rho C_p$, where $\alpha$ is the thermal diffusivity and measured by a LFA427 Microflash (NETZSCH-Gerätebau GmbH, Germany), $C_p$ is the specific heat capacity and measured by Differential Scanning Calorimeter (DSC-200F3, NETZSCH-Gerätebau GmbH, Germany), and $\rho$ is the density of the samples and characterized by the Archimedes method[17].

### Electrical and pyroelectric properties characterization
To study the electrical properties, the samples were ground into 0.3 mm and coated with silver paste on both sides with the electrode diameter of 6 mm and then sintered at 600 °C for 15 min. After that, all

the samples were poled in silicone oil using a high-voltage power supply (ET2673D-10kV) with applied field of 5 kV mm$^{-1}$ for 20 min at room temperature. The relative permittivity and dielectric loss were measured by the Dielectric Properties Testing System (DPTS-RT-1000, Wuhan Yanhe Technology Co. Ltd., China). The Polarization-electric field (*P-E*) hysteresis loops were measured with the Ferroelectric Analyzer System (TF2000, aixACCT, Germany). The pyroelectric properties were measured using the Pyroelectric Test System (PCTS-3000, Wuhan Yanhe Technology Co. Ltd., China) for poled samples at a heating rate of 2 °C min$^{-1}$.

### Measurement of energy harvesting

The Thermal Energy Harvesting Testing System comprises a signal generator, power amplifier, Peltier cell, an electrometer, computer and display screen as shown in Supplementary Fig. 13. A signal generator and power amplifier are employed to actuate the Peltier cell to produce a temperature fluctuation of 2 °C as shown in Fig. 8a. A Keithley 6517B electrometer is utilized to record the output current and voltage with various electrical loads, and computer and screen display and store the corresponding data.

## Data availability

The authors declare that the data that support the findings of this study are available within the article and its Supplementary Information files. All other relevant data are available from the corresponding authors upon request.

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

## Acknowledgements
This work is supported by the following grants: National Natural Science Foundation of China under Grant Nos. 52102128 (M.S.), 5217211 (Q.Z.), and 52272107 (Y.C.); Natural Science Foundation of Hubei Province under Grant No. 2021CFB117 (M.S.); Open Project Fund of Hubei Key Laboratory of Ferro & Piezoelectric Materials and Devices under Grant No. K202010 (M.S.); Innovation Program of Wuhan-Shuguang under Grant No. 2022010801020329 (M.S.); A*STAR, Singapore, RIE2020 Advanced Manufacturing and Engineering (AME) Programmatic Fund under Grant No. A20G9b0135 (K.Y.).

## Author contributions
M.S., S.J., and Y.C. conceived the idea of this work. K.L. fabricated the BNT-based pyroelectric ceramics. M.S. and K.L. carried out the measurements of microstructure, electrical and pyroelectric properties. G.-h.Z. performed the calculation of the phonon-dispersion, DOS and phonon group velocity. Q.L. performed the measurement of thermal parameters. Q.Z. and G.-h.Z. carried out the measurement of energy harvesting. M.S., H.Z., and S.J. analyzed all the data and prepared the manuscript. G.-z.Z. and K.Y. revised the manuscript. All authors discussed the results and commented on the paper. S.M., Y.C., Q.Z., and K.Y. supervised the research.

## Competing interests
The authors declare no competing interests.
