## [Peer Review File · Nature Communications]

Thermoelectric coupling effect in BNT-BZT-xGaN pyroelectric ceramics for low-grade temperature driven energy harvestingREVIEWER COMMENTS

Reviewer #1 (Remarks to the Author):

Pyroelectric energy harvesters have received increasing attention in recent years due to their ability to generate electric energy during temperature fluctuations. This manuscript reported a high-performance hybrid BNT-BZT-xGaN pyroelectric materials with improved thermal conductivity and pyroelectric coefficient. The following issues should be clarified before publication.

1. The expressions of “temperature fluctuation” and “temperature gradient” should be defined and clarified. In lines 331, 344, 350, 354, et al., the “temperature gradient” was used, while in line 329, “temperature fluctuation” was used. Is it the temperature gradient or the temperature fluctuation that causes the pyroelectric effect?
2. The presentation of the manuscript needs to be more accurate, concise, and logical. For example, in line 210, the conclusion “thus, ... not only gives rise to the enhancement of the lattice heat conduction but also improves the spontaneous polarization of the samples.” appears before the reason “Because the spontaneous polarization of BNT-based ceramics”
3. The term “the dipole pinning effect” in Fig. 7d seems not defined in the manuscript. In addition, the dipoles of BNT-BZT-xGaN were consistently oriented in the same direction after heating in Fig. 7d. The cause of this phenomenon was not mentioned in the manuscript.
4. It should be clearly stated in the figure or the figure caption whether the data are from theoretical calculations, simulations, or experiments.

Reviewer #2 (Remarks to the Author):

This work provides a valuable guidance for designing pyroelectric materials with further improved thermal energy harvesting performance. It is interesting to design a hybrid system comprising lead-free BNT-BZT pyroelectric matrix and high thermal conductivity GaN as dopant. There is a lot of research on the mechanism of the developed materials in this article, but the research on their pyroelectric properties is insufficient. Some other aspect should also be improved as detailed below.

1. The title should be revised, the “Thermoelectric coupling effect” is less elaborated in the paper, while the keyword “GaN” is not reflected in the title.
2. Line 62, “almost all of them are achieved by increasing the p or/and dT/dt ”. Is this statement (or/and) appropriate?
3. “0.12 μA and an open-circuit peak voltage of 58 V driven by a low-grade temperature difference of 2 $^{\circ}\text{C}$ ” 0.12 μA may not be the maximum value and will result in a larger short-circuit current as the load resistance decreases.
4. The content shown in Fig.1 is not fully described in the main text and should be supplemented.
5. In section 2.5 of the paper, it is mentioned that apply silver paste before sintering. How to ensure the uniformity of the silver paste layer during the sintering process, and how to ensure the good conductivity of the silver paste after sintering?

6. The description of Fig.5a in the paper is incomplete. The reason for the downward trend of thermal conductivity should be analyzed as the content of GaN increases.
7. There is a lack of explanation in the main text regarding “the dipole pinning effect” mentioned in Fig.7d.
8. Fig.7 mentioned in lines 342 to 346 is marked incorrectly, and it should be Fig.8
9. Without a description of Fig.8d, it is impressive to obtain a current of 0.1 μ A at a 50M Ω resistance. Is it more representative to attempt to measure the peak value that the maximum short-circuit current can reach?
10. The selection of load resistance is unreasonable. The minimum limit is 10 M Ω , which is still too large. The electrical output under smaller load gradient resistors should be supplemented, such as R=10 Ω ,100 Ω ,1K Ω ,10K Ω ,100K Ω ,1M Ω ,10M Ω .
11. According to the formula 1. The pyroelectric properties of materials are directly proportional to dT/dt, so the pyroelectric current and voltage obtained at different ΔT should be added.

Response Letter (NCOMMS-23-13853A)

Reviewer #1 (Remarks to the Author):

Pyroelectric energy harvesters have received increasing attention in recent years due to their ability to generate electric energy during temperature fluctuations. This manuscript reported a high-performance hybrid BNT-BZT-xGaN pyroelectric materials with improved thermal conductivity and pyroelectric coefficient. The following issues should be clarified before publication:

1. The expressions of “temperature fluctuation” and “temperature gradient” should be defined and clarified. In lines 331, 344, 350, 354, et al., the “temperature gradient” was used, while in line 329, “temperature fluctuation” was used. Is it the temperature gradient or the temperature fluctuation that causes the pyroelectric effect?

Response:

Thank you for pointing out the unclarity caused here. The “temperature fluctuation” refers to an upward or downward temperature change, which is followed by a further change in temperature in the opposite direction, while “temperature gradient” describes the rate of the temperature changes.

In this work, it is the temperature fluctuation that causes the pyroelectric effect. As we know, the pyroelectric effect is related to the change in polarization due to the change in temperature.

In the revised manuscript, we have changed the “temperature gradient” into “temperature fluctuation” or “temperature variation”, as marked in blue on Pages 2, 15, 16, 18, 21.

2. The presentation of the manuscript needs to be more accurate, concise, and logical. For example, in line 210, the conclusion “thus, ... not only gives rise to the enhancement of the lattice heat conduction but also improves the spontaneous polarization of the samples.” appears before the reason “Because the spontaneous polarization of BNT-based ceramics”

Response:

We appreciate the careful examination by the reviewer to note the defect in our statement. In the revised manuscripts, we have corrected the statement as below:

“In addition, the phonon distribution of Ga atom is located in the region between 10 and 375 cm^{-1} , which has overlapped with the atom phonon DOS of Na, Ti, and O. These overlapped phonon distributions stand for the resonance vibration between Ga and Na, Ti, O atoms, which contributes to the enhancement of the lattice vibration of BNT-BZT. The enhancement of the lattice vibration of BNT-BZT not only gives rise to the enhancement of the lattice heat conduction, but also improves the spontaneous polarization of the samples. As the spontaneous polarization of BNT-based ceramics is prevalingly derived from the movement of Ti and O atoms, the resonance vibration among Ga and Ti, O atoms (as shown in Fig. 4(b)) improves the vibration amplitude and frequency of Ti-O (as shown in Fig. 4(f) and supplementary Video V1), which can be confirmed by the group velocity of BNT-BZT and BNT-BZT- x GaN, with $x=0.1$ wt% as shown in Fig. 4(c)-(d).” All the changes are marked in blue on Page 8.

We carefully checked through the manuscript to make it more accurate, concise, and logical.

3. The term “the dipole pinning effect” in Fig. 7d seems not defined in the manuscript. In addition, the dipoles of BNT-BZT- x GaN were consistently oriented in the same direction after heating in Fig. 7d. The cause of this phenomenon was not mentioned in the manuscript.

Response:

The dipole pinning effect typically refers to that the defect charges or dipoles act as pinning points for the motion of electric dipoles and thus provide pinning effects in electric dipoles activities. To be specific in this work, there are two types of dipoles in the BNT-BZT- x GaN ceramics. One is electric dipoles originated from relative displacement between B-site Ti ions and O ions as shown in Fig. 7(c)-(d) and the other one is defect dipoles formed by various lattice defects. During poling, both the lattice displacement electric dipoles and defect dipoles reorient along the applied electric field; when the electric field is removed, both the lattice displacement electric dipoles and defect dipoles oscillate to reach new equilibrium conditions.

The defect dipoles usually have the low migration speed of defect. Meanwhile, the defect dipoles interact with lattice displacement electric dipoles acting as pinning points for the electric dipoles motion and thus providing pinning effects in electric dipoles activities. The pinning effect can be observed from the shift of ferroelectric hysteresis (P - E) loops along the electric field axis (defined as internal bias field) and constriction of the P - E loops. As shown below, Fig. S10(a) and (c) show the P - E loops of pure BNT-BZT and BNT-BZT- x GaN with $x=0.1$ wt% at the temperature range of 30~100 °C. It can be seen from Fig. S10(a) and (c) that the P - E loops become pinched and the polarization increases with the increase in temperature. Fig. S10(b) and (d) illustrate positive coercive electric field (E_{c+}), negative coercive electric field (E_{c-}) and internal bias field (E_i) of pure BNT-BZT and BNT-BZT- x GaN with $x=0.1$ wt% at the temperature range of 30~100 °C. Here the E_i is calculated according to the equation ($E_i = (E_{c+} + E_{c-})/2$). It can be seen from Fig. S10(b) and (d) that the E_i increases after introducing GaN into BNT-BZT ceramics, which means more defect dipoles act as pinning points in BNT-BZT- x GaN with $x=0.1$ wt% to interact with the displacement electric dipoles. Therefore, BNT-BZT- x GaN with $x=0.1$ wt% have larger capacity to absorb a greater quantity of opposite electrical charge on both sides of samples in response to temperature change and provide the potential for giant pyroelectric effect.

As to the question of the dipoles were consistently oriented in the same direction after heating in Fig. 7d, we are very sorry about the inaccuracy in the illustrative drawing that causes the confusion. Actually, after heating, the dipoles in BNT-BZT- x GaN with $x=0.1$ wt% are orientated with certain level of random just like pure BNT-BZT ceramics due to the temperature fluctuations.

In the revised manuscripts, we added the explanation and improved the corresponding figures (marked in blue on Page 13).

Fig. S10 Polarization hysteresis loops measured by Ferroelectric Analyzer System. (a) and (c) The temperature dependence of P - E loops for pure BNT-BZT and BNT-BZT- x GaN with $x=0.1$ wt%. (b) and (d) The temperature dependence of E_{c+} , E_{c-} and E_i for pure BNT-BZT and BNT-BZT- x GaN with $x=0.1$ wt%.

Fig.7 Pyroelectric property measured by Pyroelectric Properties Testing System and schematic illustration for the mechanisms of pyroelectric energy harvesting. (a) The temperature dependent pyroelectric coefficient of BNT-BZT-xGaN ceramics with different content of GaN, (b) the T_{peak} , T_{FR} and p_{peak} of BNT-BZT-xGaN ceramics with $x=0-0.2$ wt%. (c) and (d) The illustration for the mechanisms of pyroelectric energy harvesting for BNT-BZT and BNT-BZT-xGaN with $x=0.1$ wt%, respectively (At a steady state ($dT/dt=0$), in comparison with BNT-BZT sample, the poled electric dipoles of BNT-BZT-xGaN with $x=0.1$ wt% ceramic absorb a greater quantity of opposite electrical charge on both sides of samples due to the enhanced dipole pinning effect, providing the potential for giant pyroelectric effect. When $dT/dt>0$, the electric dipoles in BNT-BZT-xGaN with $x=0.1$ wt% ceramic oscillate within a larger degree of alignment, leading to larger currents in the circuits compared with that of BNT-BZT samples.).

4. It should be clearly stated in the figure or the figure caption whether the data are from theoretical calculations, simulations, or experiments.

Response:

To address the concern of the reviewer, we clearly explained in the figure caption whether the data are from theoretical calculations, simulations or experiments as shown below and marked in blue in the revised manuscript.

Fig. 1 Schematic diagram of a pyroelectric-based thermoenergetic energy harvesting system fabricated by BNT-BZT-0.1 wt% GaN ceramic. The introduction of GaN facilitates the resonance vibration between Ga and Ti, O atoms and enhances the lattice vibration of TiO_6 octahedra, which not only contributes to the enhancement of the lattice heat conduction, but also improves the pyroelectric properties. This thermoelectric coupling modulated energy harvesting system can light up the LED bulb by storing the electricity in the capacitor. When the voltage of the capacitor reaches 2.2 V, the capacitor is triggered to discharge and light up the LED bulb.

Fig. 2 Compositional heterogeneity and microstructure measured by FE-SEM and EDS.

(a)-(i) The backscattering diffraction (BSD) image and the corresponding elemental distribution of BNT-BZT- x GaN with $x=0.1$ wt%.

Fig. 3 Crystal structure measured by XRD and the Rietveld refined data calculated by the GSAS-EXPGUI software. (a) The XRD patterns of BNT-BZT- x GaN ceramics with various contents of GaN, (b) the magnified XRD peaks at around 40.0° and 46.5° for BNT-BZT- x GaN samples with $x=0-0.2$ wt%, (c) the Rietveld refined lattice parameters (a , b , c , α , β , γ) and distance of Ti-O for BNT-BZT- x GaN samples with various contents of GaN, and (d) the structural images of BNT-BZT and BNT-BZT- x GaN samples, respectively.

Fig. 4 The theoretical calculation of lattice vibrations and phonon structure by CASTEP.

The phonon density of state (PDOS) of (a) BNT-BZT and (b) BNT-BZT- x GaN with $x=0.1$ wt%, and the phonon group velocity of (c) BNT-BZT and (d) BNT-BZT- x GaN with $x=0.1\%$. (e) The Raman scattering spectra of BNT-BZT- x GaN with $x=0-0.2$ wt% measured by laser

Raman spectrometer. (f) The vibration frequency of Ti-O for BNT-BZT and BNT-BZT-xGaN with $x=0.1$ wt%, respectively.

Fig. 5 Thermal transport. (a) The temperature-dependent thermal conductivity of BNT-BZT-xGaN ceramics calculated by the equation $k = \alpha \rho C_p$, and (b) the simulation of temperature distributions for the pristine and BNT-BZT-xGaN sample with $x=0.1$ wt% by COMSOL Multiphysics software at the same time duration.

Fig. 6 Dielectric and ferroelectric properties measured with Dielectric Properties Testing System and Ferroelectric Analyzer System. (a) The temperature dependent dielectric constant (ϵ_r) and loss ($\tan\delta$) for BNT-BZT-xGaN samples with various content of GaN in the

temperature range of 25-400 °C, (b) the T_m , T_{FR} , T_s , room temperature ϵ_r and $\tan\delta$ of BNT-BZT- x GaN samples with $x=0-0.2$ wt%, (c) the P - E hysteresis loops, and (d) P_r , P_s , and E_c of BNT-BZT- x GaN samples with various content of GaN at room temperature.

Fig.7 Pyroelectric property measured by Pyroelectric Properties Testing System and schematic illustration for the mechanisms of pyroelectric energy harvesting. (a) The temperature dependent pyroelectric coefficient of BNT-BZT- x GaN ceramics with different content of GaN, (b) the T_{peak} , T_{FR} and p_{peak} of BNT-BZT- x GaN ceramics with $x=0-0.2$ wt%. (c) and (d) The illustration for the mechanisms of pyroelectric energy harvesting for BNT-BZT and BNT-BZT- x GaN with $x=0.1$ wt%, respectively (At a steady state ($dT/dt=0$), in comparison with BNT-BZT sample, the poled electric dipoles of BNT-BZT- x GaN with $x=0.1$ wt% ceramic absorb a greater quantity of opposite electrical charge on both sides of samples due to the enhanced dipole pinning effect, providing the potential for giant pyroelectric effect. When $dT/dt>0$, the electric dipoles in BNT-BZT- x GaN with $x=0.1$ wt% ceramic oscillate within a larger degree of alignment, leading to larger currents in the circuits compared with

that of BNT-BZT samples.).

Fig. 8 Pyroelectric energy harvesting experimental results measured by the Thermal Energy Harvesting Testing System. (a) The time dependent temperature variation curves of samples. (b) and (c) The short-circuit current and open-circuit voltage of BNT-BZT-xGaN ceramics with various content of GaN. (d) and (e) The pyroelectric current and voltage

obtained at different load resistance from 0-3000 M Ω , (f) The peak current, peak voltage, and output energy density (calculated by the equation $w = \int_{t=0}^{t=T} \frac{I \times U}{S \times d} dt$) of BNT-BZT-xGaN ceramic with $x=0.1$ wt%. (g) The electrical circuit diagram of the pyroelectric energy harvesting device and the photograph of the LED lit up by the device.

In addition, we also explained in the figure captions whether the data are from theoretical calculations, simulations or experiments in the part of Supporting Information.

Reviewer #2 (Remarks to the Author):

This work provides a valuable guidance for designing pyroelectric materials with further improved thermal energy harvesting performance. It is interesting to design a hybrid system comprising lead-free BNT-BZT pyroelectric matrix and high thermal conductivity GaN as dopant. There is a lot of research on the mechanism of the developed materials in this article, but the research on their pyroelectric properties is insufficient. Some other aspect should also be improved as detailed below.

1. The title should be revised, the “Thermoelectric coupling effect” is less elaborated in the paper, while the keyword “GaN” is not reflected in the title.

Response:

According to the suggestion of the reviewer, we revised the title “Thermoelectric coupling effect in BNT-based pyroelectric ceramics for low-grade temperature driven energy harvesting” into “Thermoelectric coupling effect in BNT-BZT-xGaN pyroelectric ceramics for low-grade temperature driven energy harvesting” in the revised manuscripts and marked in blue on Page 1.

2. Line 62, “almost all of them are achieved by increasing the p or/and dT/dt ”. Is this statement (or/and) appropriate?

Response:

Thank the reviewer for pointing out the inappropriate statement. In the revised manuscripts,

we revised the statement into “some of them are achieved by increasing the p or/and dT/dt ” and marked in blue on Page 3. In addition, we carefully checked the statements in this manuscript to avoid the similar inaccuracy issue.

3. “0.12 μA and an open-circuit peak voltage of 58 V driven by a low-grade temperature difference of 2 $^{\circ}\text{C}$ ” 0.12 μA may not be the maximum value and will result in a larger short-circuit current as the load resistance decreases.

Response:

In response to the comment of the reviewer, we reduced the load resistance and measured output current and voltage of BNT-BZT- $x\text{GaN}$ sample with $x=0.1$ wt% under smaller load resistor such as 10 k Ω , 50 k Ω , 100 k Ω , 1 M Ω , 5 M Ω as shown below in Fig. 8 (d)-(e). It can be seen from Fig. 8(d)-(e) that the output current increases with decrease of load resistance, when the load is below 1 M Ω , the output peak current is approximately 0.12 μA , which is equal to the short-circuit current of 0.12 μA ; and the short-circuit peak current of 0.12 μA is the maximum value as the load resistance is zero.

To compare the influence of load resistor on pyroelectric current and voltage, we added the output current and voltage as well as the energy density of BNT-BZT- $x\text{GaN}$ sample with $x=0.1$ wt% under smaller load resistance such as 10 k Ω , 50 k Ω , 100 k Ω , 1 M Ω , 5 M Ω in Fig. 8(d)-(f). As shown in Fig.8(d)-(e), the pyroelectric current is approximately equal to the short-circuit current and pyroelectric voltage is approximately equal to zero when the load resistance is lower than 1 M Ω . When the load resistance is larger than 1 M Ω , the pyroelectric current decreases with the increase in load resistance while the pyroelectric voltage increase with the increase in load resistance. In addition, the output energy density of a 10 s period is approximately equal to zero when the load resistance is lower than 1 M Ω , When the load resistance is larger than 1 M Ω , the output energy density firstly increases and then decreases with the increase in load resistance and reaches the maximum value of 80 $\mu\text{J cm}^{-3}$ at the load resistance of 600 M Ω . The underlying reason here is the current is determined by the entire resistance in the electrical circuit including the internal resistance of the pyroelectric material, not only the load resistance. When the load resistance is significantly lower than the internal

resistance, the current would not increase substantially with further reducing the load resistance. From circuit analysis, the maximum output power occurs with the match of load and internal resistances, or accurately impedance match.

In the revised manuscripts, we added the figure and explanation and highlighted in violet on page 15-18.

Fig. 8 Pyroelectric energy harvesting experimental results measured by the Thermal Energy Harvesting Testing System. (a) The time dependent temperature variation curves of samples. (b) and (c) The short-circuit current and open-circuit voltage of BNT-BZT- x GaN ceramics with various content of GaN. (d) and (e) The pyroelectric current and voltage obtained at different load resistance from 0-3000 M Ω , (f) The peak current, peak voltage, and output energy density (calculated by the equation $w = \int_{t=0}^{t=T} \frac{I \times U}{S \times d} dt$) of BNT-BZT- x GaN ceramic with $x=0.1$ wt%. (g) The electrical circuit diagram of the pyroelectric energy harvesting device and the photograph of the LED lit up by the device.

4. The content shown in Fig.1 is not fully described in the main text and should be supplemented.

Response:

To address the concern of the reviewer, we explained the Fig. 1 in detail in the figure caption as shown below. In addition, we added the corresponding explanation below in the part of Introduction and marked in blue on Pages 3, 4: “As shown in Fig. 1, the introduction of GaN facilitates the resonance vibration between Ga and Ti, O atoms, leading to the enhancement of the lattice vibration of TiO₆ octahedra. This not only contributes to the enhancement of the lattice heat conduction, but also improves the spontaneous polarization, resulting in the simultaneous improvement of dT/dt and p . Therefore, a high thermoelectrical energy harvesting density of 80 $\mu\text{J cm}^{-3}$ at the optimal load resistance of 600 M Ω has been obtained in BNT-BZT- x GaN ceramic with $x=0.1$ wt%. In addition, this thermoelectric coupling modulated energy harvesting system can charge the capacitor and control the capacitor to discharge and light up the LED bulb when connected with a capacitor.”

Fig. 1 Schematic diagram of a pyroelectric-based thermoelectrical energy harvesting system fabricated by BNT-BZT-0.1 wt% GaN ceramic. The introduction of GaN facilitates the resonance vibration between Ga and Ti, O atoms and enhances the lattice vibration of TiO_6 octahedra, which not only contributes to the enhancement of the lattice heat conduction, but also improves the pyroelectric properties. This thermoelectric coupling modulated energy harvesting system can light up the LED bulb by storing the electricity in the capacitor. When the voltage of the capacitor reaches 2.2 V, the capacitor is triggered to discharge and light up the LED bulb.

5. In section 2.5 of the paper, it is mentioned that apply silver paste before sintering. How to ensure the uniformity of the silver paste layer during the sintering process, and how to ensure the good conductivity of the silver paste after sintering?

Response:

Here, the silver paste is a kind of high temperature firing conductive silver paste, which is fired around 500-800 °C to form a film. This type of high temperature silver paste uses glass powder or oxide as binding agent. The firing process at this type of high temperature silver paste includes softening of glass powder, infiltration of silver powder and matrix by glass liquid, rearrangement of silver powder particles driven by glass liquid and liquid phase solidification shrinkage. The firing at 600 °C for 15 min is beneficial to reach the balance

between viscosity and fluidity of the glass liquid, which has good effect on infiltrating of silver powder and rearrangement of silver powder particles. Therefore, suitable firing temperature and holding time is crucial to form uniform conductive silver paste layer. After firing process, the high temperature firing conductive silver paste is characterized by great weldability, great printability, strong adhesion and low specific resistance. The adhesion strength of this kind of high temperature firing silver paste is larger than 40 MPa and the specific resistivity is smaller than 3 $\mu\Omega$ cm. Therefore, the film formed by this kind of silver paste has good uniformity and good conductivity.

6. The description of Fig.5a in the paper is incomplete. The reason for the downward trend of thermal conductivity should be analyzed as the content of GaN increases.

Response:

We appreciate the valuable comment from the detailed examination by the reviewer. As shown in Fig. 5(a), with the introduction of GaN, the thermal conductivity first increases from 1.48 $\text{W m}^{-1} \text{K}^{-1}$ to 1.61 $\text{W m}^{-1} \text{K}^{-1}$ as the GaN content increases from 0 to 0.1 wt% at room temperature, then decreases to 1.52 $\text{W m}^{-1} \text{K}^{-1}$ when the GaN content further increases to 0.2 wt%. The decrease in thermal conductivity may be attributed to the decrease in mean-free-paths due to phonon-boundary scattering. As we know, heat is carried by phonons with a broad distribution of mean-free-paths and the thermal conductivity is proportional to mean-free-paths. As the GaN content further increases to 0.2 wt%, the grain size of BNT-BZT-xGaN ceramics decreases to 1.57 μm and introduces more grain boundaries. With the introduction of boundaries, phonons with long mean free paths interact with the boundaries and reduce their mean free paths, which narrow the distribution of mean-free-paths. Therefore, the thermal conductivity of BNT-BZT-xGaN decreases to 1.52 $\text{W m}^{-1} \text{K}^{-1}$ when the GaN content further increases to 0.2 wt%.

In the revised manuscript, we added the explanation and marked in blue on Page 10.

7. There is a lack of explanation in the main text regarding “the dipole pinning effect” mentioned in Fig.7d.

Response:

The dipole pinning effect typically refers to that the defect charges or dipoles act as pinning points for the motion of electric dipoles and thus provide pinning effects in electric dipoles activities. To be specific in this work, there are two types of dipoles in the BNT-BZT- x GaN ceramics. One is electric dipoles originated from relative displacement between B-site Ti ions and O ions as shown in Fig. 7(c)-(d) and the other one is defect dipoles formed by various lattice defects. During poling, both the lattice displacement electric dipoles and defect dipoles reorient along the applied electric field; when the electric field is removed, both the lattice displacement electric dipoles and defect dipoles oscillate to reach new equilibrium conditions. The defect dipoles usually have the low migration speed of defect. Meanwhile, the defect dipoles interact with lattice displacement electric dipoles acting as pinning points for the electric dipoles motion and thus providing pinning effects in electric dipoles activities. The pinning effect can be observed from the shift of ferroelectric hysteresis (P - E) loops along the electric field axis (defined as internal bias field) and constriction of the P - E loops. As shown below, Fig. S10(a) and (c) show the P - E loops of pure BNT-BZT and BNT-BZT- x GaN with $x=0.1$ wt% at the temperature range of 30~100 °C. It can be seen from Fig. S10(a) and (c) that the P - E loops become pinched and the polarization increases with the increase in temperature. Fig. S10(b) and (d) illustrate positive coercive electric field (E_{c+}), negative coercive electric field (E_{c-}) and internal bias field (E_i) of pure BNT-BZT and BNT-BZT- x GaN with $x=0.1$ wt% at the temperature range of 30~100 °C. Here the E_i is calculated according to the equation ($E_i = (E_{c+} + E_{c-})/2$). It can be seen from Fig. S10(b) and (d) that the E_i increases after introducing GaN into BNT-BZT ceramics, which means more defect dipoles act as pinning points in BNT-BZT- x GaN with $x=0.1$ wt% to interact with the displacement electric dipoles. Therefore, BNT-BZT- x GaN with $x=0.1$ wt% have larger capacity to absorb a greater quantity of opposite electrical charge on both sides of samples in response to temperature change and provide the potential for giant pyroelectric effect.

In the revised manuscripts, we added the explanation and marked in blue on Page 13.

Fig. S10 Polarization hysteresis loops measured by Ferroelectric Analyzer System. (a) and (c) The temperature dependence of P - E loops for pure BNT-BZT and BNT-BZT- x GaN with $x=0.1$ wt%, (b) and (d) The temperature dependence of E_{c+} , E_{c-} and E_i for pure BNT-BZT and BNT-BZT- x GaN with $x=0.1$ wt%.

8. Fig.7 mentioned in lines 342 to 346 is marked incorrectly, and it should be Fig.8

Response:

Thank the reviewer' for finding the error. In the revised manuscripts, we corrected the Fig. 7 into Fig. 8 and carefully checked all the figures to avoid the similar problem.

9. Without a description of Fig.8d, it is impressive to obtain a current of $0.1\mu\text{A}$ at a $50\text{M}\Omega$ resistance. Is it more representative to attempt to measure the peak value that the maximum short-circuit current can reach?

Response:

As we answered in question 3 above, it can be seen from Fig. in Fig.8(d)-(e) that the

pyroelectric current is approximately equal to the short-circuit current and pyroelectric voltage is approximately equal to zero when the load resistance is lower than 1 M Ω . When the load resistance is larger than 1 M Ω , the pyroelectric current decreases with the increase in load resistance while the pyroelectric voltage increase with the increase in load resistance.

10. The selection of load resistance is unreasonable. The minimum limit is 10 M Ω , which is still too large. The electrical output under smaller load gradient resistors should be supplemented, such as R=10 Ω , 100 Ω , 1K Ω , 10K Ω , 100K Ω , 1M Ω , 10M Ω .

Response:

To address the concern of the reviewer, we measured the output current and voltage as well as calculated the energy density of BNT-BZT- x GaN sample with $x=0.1$ wt% under smaller load resistance such as 10 k Ω , 50 k Ω , 100 k Ω , 1 M Ω , 5 M Ω as shown below in Fig. S11 (a)-(b). It can be seen from Fig. S11 (a)-(b) that the voltage increases while the current decreases with the increase of load resistance; when the load resistance is below 1 M Ω , the peak voltage is below 0.15 V while the peak current is approximately 0.12 μ A, which is equal to the short-circuit current (0.12 μ A). It indicates that the result is similar to short circuit when the load resistance is below 1 M Ω due to the large difference between the load resistor and resistance of BNT-BZT- x GaN sample with $x=0.1$ wt%. The underlying reason here is the current is determined by the entire resistance in the electrical circuit including the internal resistance of the pyroelectric material, not only the load resistance. When the load resistance is significantly lower than the internal resistance, the current would not increase substantially with further reducing the load resistance. From circuit analysis, the maximum output power occurs with the match of load and internal resistances, or accurately impedance match.

To compare the influence of load resistance on pyroelectric current and voltage, we added the output current and voltage as well as the energy density of BNT-BZT- x GaN sample with $x=0.1$ wt% under smaller load resistor such as 10 k Ω , 50 k Ω , 100 k Ω , 1 M Ω , 5 M Ω in Fig. 8(d)-(f). As shown in Fig.8(d)-(e), the pyroelectric current is approximately equal to the short-circuit current and pyroelectric voltage is approximately equal to zero when the load resistance is lower than 1 M Ω . When the load resistance is larger than 1 M Ω , the pyroelectric

current decreases with the increase in load resistance while the pyroelectric voltage increase with the increase in load resistance. In addition, the output energy density of a 10 s period is approximately equal to zero when the load resistance is lower than 1 M Ω , When the load resistance is larger than 1 M Ω , the output energy density firstly increases and then decreases with the increase in load resistance and reaches the maximum value of 80 $\mu\text{J cm}^{-3}$ at the load resistance of 600 M Ω .

In the revised manuscripts, we added the figure and explanation (highlighted in blue on Pages 15-18).

Fig. S11 Pyroelectric energy harvesting experimental results measured by the Thermal Energy Harvesting Testing System. (a) and (b) The pyroelectric current and voltage obtained at different load resistances from 10 k Ω to 5 M Ω .

11. According to the formula 1. The pyroelectric properties of materials are directly proportional to dT/dt , so the pyroelectric current and voltage obtained at different ΔT should be added.

Response:

To address the concern of the reviewer, we measured short-circuit current and open-circuit voltage of BNT-BZT-xGaN ceramic with ΔT at 1 °C, 3 °C and 4 °C, respectively. As shown below in Fig. S12(a)-(c), both short-circuit current and open-circuit voltage show similar

trend of increasing first and then decreasing with further increase in GaN content, and the short-circuit current and open-circuit voltage reach the maximum value in BNT-BZT-xGaN ceramic with $x=0.1$ wt% for all the cases of ΔT of 1 °C, 3 °C or 4 °C. In addition, Fig. S12(d)-(e) compare the peak short-circuit currents and peak open-circuit voltages of BNT-BZT-xGaN ceramics with ΔT at 1 °C, 2 °C, 3 °C, and 4 °C, respectively. It can be seen from Fig. S12(d)-(e) that both the peak short-circuit current and peak open-circuit voltage increase with ΔT .

In the revised manuscripts, we added the figure (Fig. S12(a)-(e)) and corresponding explanation (marked in blue on Page 15).

Fig. S12 Pyroelectric energy harvesting experimental results measured with the Thermal Energy Harvesting Testing System. (a)-(c) The short-circuit current and open-circuit voltage

for BNT-BZT- x GaN ceramics with various contents of GaN and ΔT of 1 °C, 3 °C, and 4 °C, respectively. (d)-(e) The comparison of the peak short-circuit current and peak open-circuit voltage values for BNT-BZT- x GaN ceramics with ΔT of 1 °C, 2 °C, 3 °C, and 4 °C, respectively.

REVIEWERS' COMMENTS

Reviewer #1 (Remarks to the Author):

The author has made a detailed reply and necessary modifications to the review comments. This manuscript can be published on Nature Communications.

Reviewer #2 (Remarks to the Author):

The author has provided explanations for the questions raised and supplemented some information to improve the research work.